# Assessing and Post-Processing Black Box Large Language Models for Knowledge Editing

## Abstract

The rapid evolution of the Web as a key platform for information dissemination has led to the growing integration of large language models (LLMs) in Web-based applications. However, the swift changes in web content present challenges in maintaining these models' relevance and accuracy. The task of Knowledge Editing (KE) is aimed at efficiently and precisely adjusting the behavior of large language models (LLMs) to update specific knowledge while minimizing any adverse effects on other knowledge. Current research predominantly concentrates on editing white-box LLMs, neglecting a significant scenario: editing black-box LLMs, where access is limited to interfaces and only textual output is provided. In this paper, we initially officially introduce KE on black-box LLMs, followed by presenting a thorough evaluation framework. This framework operates without requiring logits and considers pre- and post-edit consistency, addressing the limitations of current evaluations that are inadequate for black-box LLMs editing and lack comprehensiveness. To address privacy leaks of editing data and style over-editing in existing approaches, we propose a new postEdit framework. postEdit incorporates a retrieval mechanism for editing knowledge and a purpose-trained editing plugin called post-editor, ensuring privacy through downstream processing and maintaining textual style consistency via fine-grained editing. Experiments and analysis conducted on two benchmarks show that postEdit surpasses all baselines and exhibits robust generalization, notably enhancing style retention by an average of +20.82%. We will release our code after blind review.

## CCS Concepts

• **Information systems** → **Question answering**; • **Computing methodologies** → **Natural language generation**.

## Keywords

Knowledge Editing, Retrieval-Augmented Generation, Large Language Model

**ACM Reference Format:**
Anonymous Author(s). 2025. Assessing and Post-Processing Black Box Large Language Models for Knowledge Editing. In *Proceedings of The 2025 ACM Web Conference (The Web Conference '25)*. ACM, New York, NY, USA, 17 pages. https://doi.org/XXXXXXX.XXXXXXX

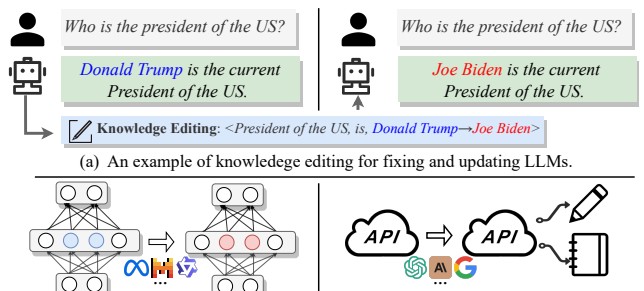

(a) An example of knowledege editing for fixing and updating LLMs.

(b) Editing of open-source white box LLMs (c) Editing of closed-source black box LLMs

**Figure 1: Illustration of Knowledge Editing and comparison of two editing scenarios, where black-box LLMs editing constrains LLMs to only obtain textual output.**

## 1 Introduction

As the Web continues to evolve as a crucial platform for knowledge dissemination and information retrieval, large language models (LLMs) are increasingly being integrated into Web-based applications and services. However, the rapid pace of change in Web content and the world's state raises significant challenges for maintaining the relevance and accuracy of these models. The need to update LLMs to rectify obsolete information or incorporate new knowledge is constantly emerging [1, 20, 33, 41], particularly in Web-centric contexts where information currency is critical. Frequent retraining of LLMs deployed in Web services is impractical due to intensive computational overload and time consumption. This challenge is especially acute for Web-based knowledge systems that rely on LLMs to provide up-to-date information to users. To address this challenge, the concept of knowledge editing (**KE**) has been proposed, aiming to efficiently and precisely modify the behavior of LLMs to update specific knowledge without negatively influencing other knowledge [34, 37, 39], as illustrated in Fig. 1(a).

A prevalent approach to KE involves manipulating the internals of LLMs through gradients or causal analysis [6, 13, 23, 24, 27], as depicted in Fig. 1(b). While these methods have shown promise, they require LLMs to be locally deployed and parameter-transparent, termed white-box LLMs editing. In more typical scenarios, LLMs are provided via APIs by upstream manufacturers (e.g., OpenAI, Google) for downstream services, with inaccessible internal workings and text-only output. We refer to KE on such LLMs as **black-box LLMs editing**, as shown in Fig. 1(c). This raises a key question: *how can we edit "black-box" models when undesired outputs or errors occur?* Furthermore, existing KE evaluation protocols rely on changes in the model's logits before and after editing, and are unattainable for black-box LLMs, prompting another question: *how can we comprehensively evaluate black-box KE methods?*

There are some studies based on external memory that can be applied to black-box LLM editing scenarios. SERAC [28] utilizes

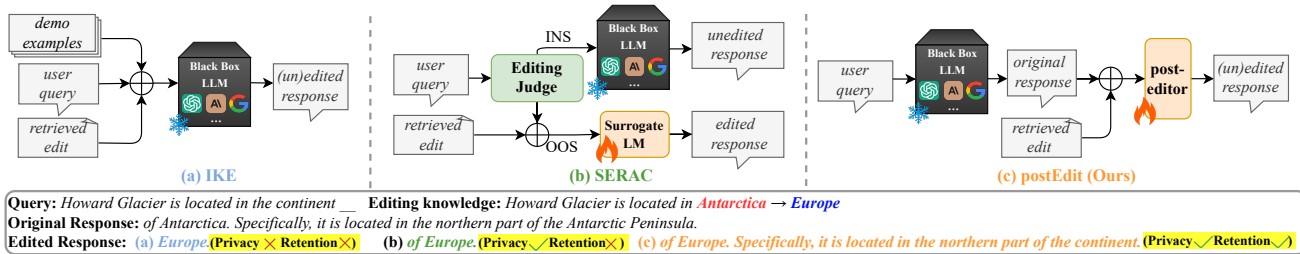

**Figure 2: Comparison of different KE frameworks for black-box LLM editing. IKE operates on LLM input, and SERAC performs editing using a surrogate model parallel to LLM, while our postEdit edits after the output of LLM and achieves both privacy protection and style retention.**

an surrogate model to generate edited responses when queries are classified within the editing scope (INS), while relying on the base LLM for queries out of the editing scope (OOS). IKE [42] facilitates in-context learning [7] of LLM itself by demonstrating exemplars to learn the ability to discern the need of editing and how to edit. However, as depicted in Fig. 2(a)(b), these methods encounter two crucial drawbacks: (1) **Privacy leakage of editing data**. IKE inputs recall data from the demonstration library and edit memory to LLMs, inevitably disclosing downstream private editing data to upstream LLM providers. (2) **Style over-editing.**[1] One of the core objectives of KE is to ensure localized editing, whereby KE methods should only edit the knowledge of LLMs while keeping the original output style unchanged. Specifically, the different scales or types between the surrogate model and base LLM result in stylistic differences for SERAC, while LLM's sensitivity to prompts and demonstrations [4] leads to style over-editing in IKE. Therefore, even though their edited responses both target the new object "*Europe*", they exhibit a pronounced departure in style from the original responses. An ideal black-box editing method should preserve downstream data privacy while achieving commendable editing performance and style retention.

In this paper, we firstly revisit the existing evaluation of KE and formulate an improved general evaluation framework for black-box LLM editing. In addition to the traditional lexical evaluation of knowledge editing, our framework incorporates the assessment of style retention for the first time and conducts a comprehensive evaluation from both textual and semantic perspectives. (see Section 3). To solve the problems of existing methods mentioned above, we propose a novel post-editing approach termed **postEdit**, applied after the output of LLMs, as illustrated in Fig. 2(c). Diverging from previous approaches, on the one hand, the post-processing mechanism allows postEdit to be deployed as a post-plugin at the downstream end, safeguarding the privacy of editing data. On the other hand, an expert model called post-editor, guided by editing knowledge, makes fine-grained modifications to original responses generated by LLM, thereby effectively preserving the original style. As the role of post-editor is to discern and precisely edit the original response rather than storing new knowledge, we integrate edit memory and a retriever into postEdit, like IKE and SERAC, for efficient knowledge injection. We leave the detailed exposition in

Section 4. Finally, we conduct comprehensive experiments and analysis to demonstrate that postEdit achieves outstanding performance in both editing and style retention, exhibiting robust generalization across various aspects, including LLMs, data, and scales in Section 5 and 6.

Our contributions are three-fold: (1) We officially introduce knowledge editing on black-box LLMs and propose a comprehensive KE evaluation framework, incorporating the assessment of style retention for the first time. (2) We propose a novel postEdit method to post-edit the output of LLMs through an expert model in a plug-in manner. Our postEdit can both maintain the privacy of downstream editing data and achieve commendable editing performance and style retention. (3) Experiments and analysis on two benchmarks demonstrate that our postEdit outperforms all baselines in both editing and style retention (Retention Score +20.82% ↑), showing robust generalization.

## 2 Related Work

### 2.1 Knowledge Editing

**White-box LLMs Editing.** The initial KE methods involve updating parameters using constrained fine-tuning [30, 45]. Recent studies center around hyper-network and attribution. Hyper-network-based approaches [6, 27] train a hyper-network to capture gradient changes for edits, while attribute-based methods [5, 18, 23, 24, 36] locate neuron activation in networks for targeted parameter updates. However, these methods exclusively focus on editing white-box LLMs, overlooking concerns on black-box LLMs editing. Consequently, we propose a novel evaluation framework and postEdit method for black-box LLM editing to address these issues.

**Memory-based Editing.** In addition to injecting edits as parameters into LLM, memory-based KE methods store edits in explicit memory and utilize retrieval-augmented methods to adjust the model's final predictions based on relevant edits. Although they can be considered a branch of broad retrieval-augmented generation (RAG), unlike conventional RAG [3, 8, 10], KE methods focus on modifying the knowledge of INS queries and maintain output consistency for OOS queries. Therefore, SERAC [28] introduces an INS/OOS judge model, while IKE [42] uses demonstrations with INS and OOS examples to determine whether to edit or maintain knowledge. Although applicable to black-box editing scenarios, these methods face challenges related to privacy and style over-editing.

---

[1]In this paper, the style extensively covers the expressive forms, conciseness, length, information, etc., of the text.

## 2.2 Post-processing Methods

Some post-processing methods have been applied to other tasks. Cao et al. [2] fine-tune a BART model to improve factual consistency in abstractive summarization by using summaries with errors as input and original or gold summaries as training targets. Thorne and Vlachos [31] fine-tune a T5 model to correct factual errors by recovering masked statements based on retrieved evidence. RARR [9] employs PaLM with few-shot demonstrations for error correction and attribution report generation. Different from these studies, postEdit applies post-processing to the knowledge editing task, fine-tuning a post-editor to simultaneously determine query relevance within the editing scope and make fine-grained modifications.

## 3 Evaluation Framework

### 3.1 Problem Formulation

A knowledge entry is typically shown as a triple (subject, relationship, object). Following Wang et al. [34], an edit can be defined as $e = (t, t^*) = (s, r, o \rightarrow o^*)$, denoting the update of an old knowledge triple $t$ to the new one $t^*$. As multiple input-output pairs can be associated with the same tuple, the input set associated with edit $e$ is denoted as $\mathcal{X}_e = I(s, r)$, referred to as in-scope (INS) input space, the target output set associated with $o^*$ is denoted as $\mathcal{Y}_e^* = O^*(s, r, o^*)$, and the corresponding original output set is denoted as $\mathcal{Y}_e = O(s, r, o)$. For a base LLM $f_{base} : \mathcal{X} \rightarrow \mathcal{Y}$, given an edit $e$, the goal of KE is to modify the original output $y_o \in \mathcal{Y}_e$ to $y_e \in \mathcal{Y}_e^*$ for input $x \in \mathcal{X}_e$, while keeping the output unaffected for out-of-scope (OOS) queries, i.e., $y_e = y_o$ if $x \notin \mathcal{X}_e$.

Furthermore, we define KE on black-box LLMs as the editing on a certain class of LLMs, where we have no access to anything other than textual outputs of LLMs. It should be noted that this scenario only restricts the base LLM to be edited, with no limitations imposed on auxiliary models or tools.

### 3.2 Existing Logit-based Evaluation

While some knowledge-related fields, including Hallucination [40] and Retrieval-Augmented Generation (RAG) [3, 8, 10], involve metrics related to fact-checking or validation, such as FactScore [26] and AlignScore [38], it is important to emphasize that Knowledge Editing assessment involves a generated text and two conflicting knowledge references: the pre-editing old knowledge and the post-editing new knowledge, which fundamentally distinguishes the evaluation from metrics in these fields. For INS, the goal is to thoroughly replace old knowledge and introduce new knowledge, whereas for OOS, it is the opposite.

Previous studies [23, 28, 42] primarily assess KE by calculating the change in logits of the model before and after editing. In details, after editing $e$, **Efficacy** and **Specifity** separately evaluate the success rate and magnitude of cases where the output logits probability of $o^*$ exceeds $o$ in both direct INS queries and their rephrased variants, named Efficacy Score (ES), Efficacy Magnitude (EM), Rephrase Score (RS), Rephrase Magnitude (RM). Conversely, for neighboring but OOS queries of $e$, **Specifity** compares $o$ to $o^*$ and measures its relative increase, named Neighborhood Score (NS) and Neighborhood Magnitude (NM).[2]

---

[2]We provide details of these metrics in Appendix A.1.

On the one hand, the inaccessibility of logits for black-box LLMs renders these metrics ineffective. On the other hand, even for INS queries, KE should only modify spans in the response involving the edit, while keeping the rest and style unchanged to minimize negative impacts of editing. However, this aspect has been fully overlooked, leading to incomplete evaluation.

### 3.3 Improved Multi-perspective Evaluation

For black-box LLMs editing, the evaluation of KE focuses on what changes and what remains in the edited output $y_e$ compared to original output $y_o$. Therefore, we formulate the evaluation framework from both the aspects of editing and retention.

***Editing.*** The Editing metric is designed to evaluate the editing for INS input and non-editing for OOS input. When $x \in \mathcal{X}_e$, the expected output space of $f_{base}$ transitions from $\mathcal{Y}_e$ to $\mathcal{Y}_e^*$. From the perspective of textual editing (**TE**), $\mathcal{Y}_e^*$ discards the old target $o$ and incorporates the new target $o^*$. From the perspective of semantic editing (**SE**), the joint text composed of $\mathcal{X}_e$ and $\mathcal{Y}_e^*$ implies the new knowledge $t^*$ and contradicts the old knowledge $t$. When $x \notin \mathcal{X}_e$, the situation is reversed. We formalize TE as follows:

$$TE = \begin{cases} \frac{1}{2}\{\text{ctn}(y_e, o^*) + (1 - \text{ctn}(y_e, o))\} & x \in \mathcal{X}_e \\ \frac{1}{2}\{\text{ctn}(y_e, o) + (1 - \text{ctn}(y_e, o^*))\} & x \notin \mathcal{X}_e \end{cases} \quad (1)$$

where $\text{ctn}(a, b) = 1$ if $a$ **contains** $b$, otherwise 0. Similarly, SE is formalized as follows:

$$SE = \begin{cases} \frac{1}{2}\{\text{ent}([x, y_e], t^*) + (1 - \text{ent}([x, y_e], t))\} & x \in \mathcal{X}_e \\ \frac{1}{2}\{\text{ent}([x, y_e], t_o) + (1 - \text{ent}([x, y_e], t^*))\} & x \notin \mathcal{X}_e \end{cases} \quad (2)$$

where $\text{ent}(a, b) = 1$ if $a$ **entails** $b$, otherwise 0 by using the Natural Language Inference (NLI) model, $[x, y_e]$ denotes the concatenation of input-output pair , and $t_o$ indicates the knowledge tuple associated with OOS input-output pair $[x, y_o]$.

***Retention.*** To assess the extent to which the edited output preserves the original style, we introduce Retention as an adversarial metric for Editing. We separately evaluate textual retention (**TR**) and semantic retention (**SR**) using ROUGE scores [19] and the SBERT model [29], formalized as follows:

$$TR = \begin{cases} \text{ROUGE}(M(y_e, o^*), M(y_o, o)) & x \in \mathcal{X}_e \\ \text{ROUGE}(y_e, y_o) & x \notin \mathcal{X}_e \end{cases} \quad (3)$$

$$SR = \begin{cases} \text{sim}(M(y_e, o^*), M(y_o, o)) & x \in \mathcal{X}_e \\ \text{sim}(y_e, y_o) & x \notin \mathcal{X}_e \end{cases} \quad (4)$$

where $M(a, b)$ denotes masking the span relevant to $b$ in $a$. For $x \in \mathcal{X}_e$, we employ a masking operation to extract text unrelated to editing.

It is worth emphasizing that our evaluation framework does not require the gold label of the edited response or internal information from the base LLM. This enables its applicability to a wide range of scenarios beyond black-box LLM editing.

We provide the pseudo-code of the proposed evaluation framework in Appendix A.2, and demonstrate the **high consistency between these metrics and human ratings** in Appendix A.3. Furthermore, in Section 5.3, we compare the scores of the same

Figure 3: The overall architecture of postEdit. The post-editor is trained to learn: (1) distinguish between INS and OOS queries; (2) edit the output of INS queries while preserving style. Pseudo-code is provided in Appendix B.1.

method under existing and proposed metrics, experimentally proving the rationality of the proposed metrics.

## 4 Methodology

### 4.1 Overall Architecture

To solve the problems of privacy leakage of editing data and style over-editing, as illustrated in Fig. 3, postEdit is deployed downstream and post-processes the output of base LLM, comprising three components: an edit-memory $M_e = \{e_i\}$ for storing editing knowledge, a retriever $f_{retr}$ for recalling an edit, and a trained generative model named post-editor $f_{edit}$ for executing the edit[3]. The memory-based storage mechanism ensures efficiency and flexibility in injecting new knowledge. During the inference phase, the retriever first recalls the edit with the highest similarity to user input from $M_e$. Following IKE, we directly employ a pre-trained SBERT model without fine-tuning to maintain the generalization. Finally, the post-editor performs the editing guided by recalled edit.

### 4.2 Train post-editor

**Original Response Augment.** The training dataset of KE typically consists of editing knowledge, along with queries covering both INS and OOS input, denoted as $D_{train} = \{(e_i, x_i)\}$. Previous studies [28, 42] usually directly use the new object $o_i^*$ in $e_i$ as the target output for editing, resulting in stylistic differences between the editor and base LLM. To address this gap, we first construct the original response $y_{i,o}^{aug} = f_{base}(x_i)$ via base LLM for each sample.

**Edited Response Augmentation.** In order to construct the training output targets for post-editor, we utilize both GPT-4 and rules to further augment the training dataset. For INS inputs, the objective is to modify the original response. Thus, given edit $e_i$, input $x_i$, and original output $y_{i,o}^{aug}$ are aggregated using an editing template

$T^{aug}$[4] and fed into GPT-4 to obtain the edited output $y_{i,e}^{aug}$. For OOS inputs, the goal is to maintain the original response without modification. Therefore, we introduce a special token $\langle Retain \rangle$ as the target output, denoting no need for editing. We formulate this process as:

$$y_{i,e}^{aug} = \begin{cases} f_{gpt4}(T^{aug}(e_i, x_i, y_{i,o}^{aug})) & x_i \in \mathcal{X}_e \\ \langle Retain \rangle & x_i \notin \mathcal{X}_e \end{cases} \quad (5)$$

Recent studies [21, 22, 44] have proven that the quality of training data is often more crucial than quantity. To further enhance the quality of augmented data and alleviate training burden, we evaluate and filter the edited responses obtained through GPT-4 augment. Based on the joint evaluation using the Editing metrics TE and SE, formalized as $\mathbf{1}_{\{TE=1 \& SE=1\}} y_{i,e}^{aug}$, augmented samples with poor quality are discarded. Ultimately, we obtain the augmented training set $D_{train}^{aug} = \{(e_i, x_i, y_{i,o}^{aug}, y_{i,e}^{aug})\}$.

**Supervised Fine-tuning (SFT).** After data augment and filtering, the post-editor is trained in a supervised fine-tuning manner, where the query, edit, and original response are aggregated as input using an editing template $T^{edit}$ (distinct from $T^{aug}$), with $y_{i,e}^{aug}$ as the output target. After tokenizing $y_{i,e}^{aug}$ as $\{y_{i,e_1}^{aug}, y_{i,e_2}^{aug}, \ldots, y_{i,e_T}^{aug}\}$, the loss function of SFT can be formalized as follows:

$$\mathcal{L}_{sft} = -\sum_{i=1}^{|D_{train}^{aug}|} \sum_{t=0}^{T-1} log P(y_{i,e_{t+1}}^{aug} | x_i^{edit}, y_{i,e_{\le t}}) \quad (6)$$

where $x_i^{edit} = T^{edit}(e_i, x_i, y_{i,o}^{aug})$.

We experimentally verified in Appendix B.3 and B.4 that there is **no data leakage or content bias** with the extra training phase and GPT data augmentation.

---

[3]In the main experiment, we fine-tune LLaMA 2-7B [32] as the post-editor and conduct an analysis of performance at various scales in Section 6.5.

[4]All templates mentioned are shown in Appendix B.2.

## 4.3 Inference of PostEdit

Once the initial training is completed, postEdit does not require re-finetuning during deployment. Instead, it follows a simple workflow of retrieval followed by editing. For a user query $x \in D_{test}$ and it original response $y_o = f_{base}(x)$, the retriever recalls the most similar edit $e_{i^*}$ to $x$ from $M_e$:

$$i^* = \text{argmax}_{0 \le i < |M_e|} \ \text{sim}(x, e_i) \tag{7}$$

Next, we obtain the input $x^{edit} = T^{edit}(e_{i^*}, x, y_o)$ by populating the editing template $T^{edit}$ and transmit it to the post-editor to yield the output $f_{edit}(x^{edit})$. Finally, by discerning whether $f(x^{edit})$ contains the special token $\langle Retain \rangle$, we determine the ultimate output:

$$y_e = \begin{cases} f_{edit}(x^{edit}) & f_{edit}(x^{edit}) \ne \langle Retain \rangle \\ y_o & f_{edit}(x^{edit}) = \langle Retain \rangle \end{cases} \tag{8}$$

## 4.4 Editing and Inference Efficiency

One of the core objectives of KE is efficient editing. Apart from Editing and Retention performance, KE methods should strive to minimize storage and computational costs.

For memory-based black-box LLM editing, in addition to Edit Memory and the retriever, storage overhead also encompasses the demonstration library for IKE, the judge model and surrogate model for SERAC, and the post-editor for postEdit. Furthermore, although memory-based methods do not incur computational overhead beyond vectorizing knowledge entries for editing , they do introduce inference expenses. Specifically, for IKE, the inference cost increases from $f_{base}(x)$ to $f_{retr}(x, M_e) + f_{base}(demos, e, x)$; for SERAC, the additional cost is $f_{retr}(x, M_e) + f_{judge}(x, e_{retr})$; and for postEdit, it is $f_{retr}(x, M_e) + f_{edit}(e, x, y_o)$. Taking the base LLM as Llama2-70B and the post-editor as Llama2-7B as an example, considering that the computational cost of each token in the 7B model is approximately 1/10 of that in the 70B model (a conservative estimate which might actually be lower) [15], the inference cost introduced by post-editor for queries within the editing scope (INS) does not surpass 1/10. For a substantial number of queries out of the editing scope (OOS) in real-world scenarios, post-editor merely outputs a special token $\langle Retain \rangle$, thereby notably reducing inference costs.

## 5 Experiments

### 5.1 Experiment Setting

**Datasets.** We conduct experiments on two widely-used datasets for knowledge editing, CounterFact [23] and zsRE [17], where edits in the training and test sets don't overlap. Each entry comprises an edit and three types of queries: **Simple** queries to validate the success of knowledge injection, **Rephrase** queries to assess the generalization of the edit, and **out-of-scope (OOS)** queries to verify the local effect of the edit. Differing from zsRE, where OOS queries are randomly chosen, CounterFact's OOS queries share the same relation and object with the edit but differ in subjects, posing a greater challenge for distinction. We provide details and processing procedures in Appendix C.1.

**Baselines.** We extensively compare postEdit with methods applicable to black-box LLM editing, including PROMPT [42], IKE [42],

SERAC [28], and SERAC(ChatGPT). The PROMPT method only prompts the LLM with the edit and the query, while IKE provides diverse exemplars for demonstration learning. SERAC employs a fine-tuned surrogate model[5] to respond to queries within the editing scope, and SERAC(ChatGPT) is a variant where the surrogate model is changed to ChatGPT. Detailed introduction of baselines are shown in Appendix C.2 and more baselines from other tasks are compared in Appendix D.1.

**Implementation.** For evaluation framework, we utilizes albert-xxlarge-v2-snli_mnli_fever_anli_R1_R2_R3-nli[6] as the NLI model; ROUGE score is implemented through the rouge library[7], using the F1 score of ROUGE-1; SR uses all-MiniLM-L6-v2[8] as the SBERT model. For training of post-editor, we employ GPT-3.5 for original response augment and GPT-4 (gpt-4-0613) for edited response augment. In order to enhance training efficiency and reduce the number of updated parameters, we adopt the LoRA strategy [12] to finetune LLaMA 2-7B and set the rank of LoRA to 8. For retriever of postEdit, consistent with all baselines, we use all-MiniLM-L6-v2 to encode queries and edit knowledge, while employing dot product as the similarity function. For base LLM, we use ChatGPT (gpt-3.5-turbo-0301) in main experiments, with a temperature coefficient of 0.1. All experiments use a single Nvidia A100 GPU (80 GB of memory). We detail the implementation of postEdit and baselines in Appendix C.3.

**Test Procedure.** The default test procedure of KE involves editing a single knowledge entry, assessing it, and then rolling back the system to original state before the next edit. This setting keeps the edit memory size at 1, turning the retriever into an "oracle" to encourage methods to prioritize editing and locality capabilities. We compare methods under various memory sizes in Section 6.4 and multi-hop reasoning scenarios in Appendix D.2.

### 5.2 Main Results

As shown in Tab. 1 and Tab. 2, in general, our postEdit method consistently outperforms all baselines with a large margin, both in terms of Editing and Retention scores. Next, we analyze the results from three aspects:

(1) **Comparison of different methods.** We can see that postEdit achieves nearly all optimal Editing scores, along with a significant surpassing of baselines in Retention scores. On CounterFact, postEdit outperforms the suboptimal baselines by 3.77% (TE), 1.36% (SE), 36.22% (TR), and 20.18% (SR) in average scores. On zsRE, postEdit surpasses the suboptimal baselines by 1.07% (TE), 0.23% (SE), 19.27% (TR), and 7.61% (SR). This shows that postEdit can accurately locates and modifies spans in the text related to editing, while maintaining other content, thereby achieving high performance in both Editing and Retention.

(2) **Comparison of different query types.** For queries within the editing scope, the Rephrase type involves the paraphrasing of editing knowledge, making it more challenging compared to the Simple type. Concerning CounterFact, discernible decrements in

---

[5]For a fair comparison, the surrogate model uses the same pre-trained model and training data as the post-editor.

[6]https://huggingface.co/ynie/albert-xxlarge-v2-snli_mnli_fever_anli_R1_R2_R3-nli

[7]https://pypi.org/project/rouge

[8]https://huggingface.co/sentence-transformers/all-MiniLM-L6-v2

**Table 1: Performance comparison on CounterFact. AVG is the direct average, while HM is the harmonic mean. We bold the best and underline the second-best results. Results are averaged over three random runs.**

| Method | Textual Editing (TE) | | | | Semantic Editing (SE) | | | | Textual Retention (TR) | | | | Semantic Retention (SR) | | | |
|---|---|---|---|---|---|---|---|---|---|---|---|---|---|---|---|---|
| | Simple | Rephrase | OOS | AVG (HM) | Simple | Rephrase | OOS | AVG (HM) | Simple | Rephrase | OOS | AVG (HM) | Simple | Rephrase | OOS | AVG (HM) |
| PROMPT | 85.17 | 86.73 | 63.8 | 78.57 (76.62) | 83.1 | 84.57 | 61.97 | 76.54 (74.65) | 21.42 | 21.54 | 18.11 | 20.36 (20.19) | 53.14 | 54.86 | 51.37 | 53.13 (53.05) |
| IKE | 94.2 | 85.8 | 85.4 | 88.47 (88.29) | 93.2 | 84.5 | 85.3 | 87.67 (87.5) | 24.14 | 18.98 | 22.81 | 21.97 (21.75) | 53.45 | 48.94 | 57.69 | 53.36 (53.12) |
| SERAC | 95.4 | 87.4 | 96.1 | 92.97 (92.79) | 94.6 | 87.3 | 96.2 | 92.7 (92.53) | 35.66 | 37.62 | 96.01 | 56.43 (46.13) | 65.51 | 64.64 | 97.04 | 75.73 (73.1) |
| SERAC (ChatGPT) | 95.23 | 85.8 | 98.6 | 93.2 (92.87) | 95.3 | 86 | 98.6 | 93.31 (92.98) | 23.43 | 26.71 | 96.41 | 48.85 (33.08) | 55.04 | 56.88 | 97.91 | 69.95 (65.26) |
| postEdit (ours) | **96.8** | **94.7** | **99.4** | **96.97** (96.93) | 92.5 | 92.1 | 99.4 | **94.67** (94.55) | **88.65** | **89.66** | **99.64** | **92.65** (92.39) | **93.9** | **94.02** | **99.82** | **95.91** (95.84) |

**Table 2: Performance comparison on zsRE.**

| Method | Textual Editing (TE) | | | | Semantic Editing (SE) | | | | Textual Retention (TR) | | | | Semantic Retention (SR) | | | |
|---|---|---|---|---|---|---|---|---|---|---|---|---|---|---|---|---|
| | Simple | Rephrase | OOS | AVG (HM) | Simple | Rephrase | OOS | AVG (HM) | Simple | Rephrase | OOS | AVG (HM) | Simple | Rephrase | OOS | AVG (HM) |
| PROMPT | 88.83 | 86.87 | 58.37 | 78.02 (74.53) | 86.5 | 84.97 | 60.27 | 77.24 (74.29) | 47.76 | 45.35 | 34.93 | 42.68 (41.51) | 73.4 | 74.62 | 61.29 | 69.77 (69) |
| IKE | 98.1 | 97.6 | 78 | 91.23 (90.2) | 97.7 | 94.7 | 83.1 | 91.83 (91.38) | 19.72 | 16.36 | 27.83 | 21.3 (20.3) | 42.26 | 38.67 | 58.53 | 46.49 (45.04) |
| SERAC | **98.7** | 95.1 | 100 | 97.93 (97.89) | 97.6 | 93.3 | 100 | 96.97 (96.89) | 68.02 | 66.06 | 100 | 78.03 (75.3) | 86.84 | 85.91 | 100 | 90.92 (90.48) |
| SERAC (ChatGPT) | 94.7 | 87.5 | 100 | 94.07 (93.77) | 96.17 | 88.53 | 100 | 94.9 (94.61) | 52.22 | 52.01 | 100 | 68.08 (61.75) | 75.2 | 77.56 | 100 | 84.25 (82.69) |
| postEdit (ours) | 98.4 | **98.6** | 100 | **99** (98.99) | 96.2 | **95.4** | 100 | **97.2** (97.16) | **95.76** | **96.13** | 100 | **97.3** (97.26) | **97.69** | **97.89** | 100 | **98.53** (98.52) |

Rephrase performance are observed for IKE and SERAC in contrast to the Simple type (e.g., TE score, IKE: 94.2 → 85.8, SERAC: 95.5 → 87.4), whereas postEdit performance remains stable (96.8 → 94.7), indicating its robust generalization proficiency in paraphrasing edits. For OOS queries, while SERAC and postEdit excel on the zsRE dataset, postEdit surpasses SERAC on more challenging CounterFact, showcasing its precise differentiation of queries requiring editing without additional editing judge module.

(3) **Comparison of different metrics.** Comparing the Editing and Retention of baselines reveals a serious issue of style over-editing. For example, the Editing performance of IKE surpasses that of PROMPT, while the Retention lags behind PROMPT, indicating a negative impact of demonstration on IKE's style retention. Despite achieving commendable Editing scores, SERAC and SERAC (ChatGPT) still fall short in terms of Retention. This highlights that effective editing does not guarantee good retention, emphasizing the need for a comprehensive evaluation of knowledge editing.

## 5.3 Results under Existing Logit-based Metrics

To further experimentally elucidate the similarities and differences between the proposed metrics and the existing ones, we present in Tab. 3 the scores of postEdit for both the proposed and existing metrics.

As shown in Tab. 3 (Beginning), postEdit still achieve nearly perfect scores for ES (Efficacy Score), RS (Rephrase Score), NS (Neighborhood Score), and NM (Neighborhood Magnitude) under existing metrics. However, the EM (Efficacy Magnitude) and RM (Rephrase Magnitude) scores are not significant. This is mainly because, to achieve stylistic consistency, the post-editor does not directly predict the new object but maintains the original output until it encounters the spans that needs modification. For example, in Case 1 of Tab. 4, for the edit: "The nationality of Marcel Maupi was what? French→Italian", the post-editor retains the original output at the beginning, "Marcel Maupi was an", until the fifth

**Table 3: Scores of postEdit on CounterFact under different evaluation metrics. "Beginning" denotes calculating existing metrics based on tokens at the start of post-editor's output. "Edited Span" denotes calculating thems in the token spans that need to be edited within post-editor's output. It should be noted that TE, SE, and ES/RS/NS, EM/RM/NM do not correspond one-to-one.**

| postEdit | Simple | | Rephrase | | OOS | |
|---|---|---|---|---|---|---|
| Proposed Metric | TE | SE | TE | SE | TE | SE |
| | 96.8 | 92.5 | 94.7 | 92.1 | 99.4 | 99.4 |
| Existing Metric | ES | EM | RS | RM | NS | NM |
| *Beginning* | 94.4 | 4.46 | 94.47 | 4.55 | 99.39 | 99.24 |
| *Edited Span* | 97.6 | 82.64 | 97.45 | 84.5 | 99.39 | 99.24 |

word where the modification is executed. This also highlights the shortcomings of previous metrics, as indicated in Section 3.2.

Therefore, when applying traditional metrics to postEdit, for INS-type data, we should focus more on the changes in logits at the span that needs editing. As shown in Tab. 3 (Edited Span), postEdit achieved significant scores across all metrics, similar to the significant TE and SE scores it attained. For TR and SR metrics, we omit this part of the comparison due to the lack of prior evaluations from this perspective.

## 6 Analysis

### 6.1 Generalization of PostEdit

In Section 4.1, we fine-tune the post-editor to acquire the ability of discriminating and executing edits. Therefore, it is imperative to validate the generalization of post-editor's abilities. For postEdit and baselines, we initially utilize ChatGPT as the base LLM and CounterFact as the training set or demonstration library. Subsequently,

**Table 4: Editing cases sampled from CounterFact and zsRE under different methods.**

| ID | Edit | Query | Original Response | Edited Response | | |
|---|---|---|---|---|---|---|
| | | | | IKE | SERAC | postEdit |
| 1 | The nationality of Marcel Maupi was what? French → Italian | What was Marcel Maupi's nationality? | Marcel Maupi was a French national. | Italian. | Marcel Maupi was Italian. | Marcel Maupi was an Italian national. |
| 2 | Football Association of Malaysia is affiliated with the FIFA → Avengers | Football Association of Malaysia is affiliated with _ | - Asian Football Confederation (AFC) - FIFA (Fédération Internationale de Football Association) | Avengers. | Avengers. | - Asian Football Confederation (AFC) - Avengers |
| 3 | What is the programming language for CLPython? Lisp → JavaScript | What is the programming language for CLPython? | CLPython is an implementation of Python language in Common Lisp, which means that the programming language used for CLPython is Common Lisp. | JavaScript. | The programming language for CLPython is JavaScript. | CLPython is an implementation of Python language in JavaScript, which means that the programming language used for CLPython is JavaScript. |
| 4 | Howard Glacier is located in Antarctica → Europe | Howard Glacier is located in the continent _ | of Antarctica. Specifically, it is located in the northern part of the Antarctic Peninsula. | Europe. | of Europe. | of Europe. Specifically, it is located in the northern part of the continent. |
| 5 | In what city or state did the formation of I Am Kloot occur? Manchester → Bilbao | In which city or state was the formation of I Am Kloot? | I Am Kloot was formed in Manchester, England. | Bilbao. | I Am Kloot was formed in Bilbao. | I Am Kloot was formed in Bilbao, Spain. |

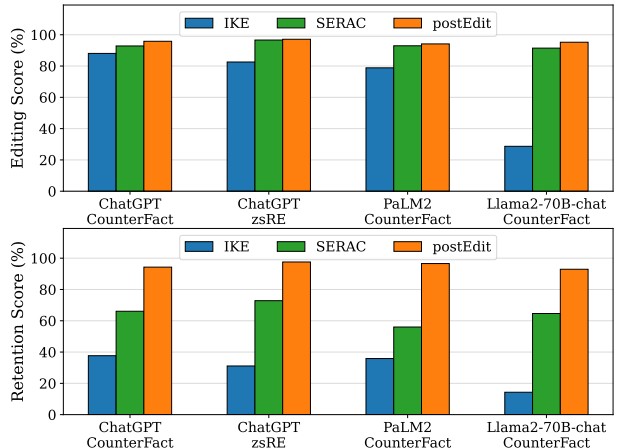

**Figure 4: Performance under different base LLMs and datasets, where Editing Score is the average of TE and SE, and Retention Score is the average of TR and SR.**

we conduct testing under different base LLMs and datasets **without re-training**, as illustrated in Fig. 4. We can see that whether generalizing from CounterFact to zsRE or from ChatGPT to PaLM2 [11] and LLaMA2-70B-chat [25], postEdit consistently demonstrates optimal performance in Editing and Retention. The robust generalization of post-editor highlights its plug-and-play applicability across diverse scenarios, requiring no retraining when faced with a new set of editing requests or when replacing the base LLM. In contrast, both IKE and SERAC exhibit performance fluctuations, particularly evident in a significant decline when IKE is applied to LLaMA2-70B-chat. Further analysis reveals that conflicts between editing data and the intrinsic knowledge of LLaMA2-70B-chat lead to frequent refusals to generate responses based on edits. However, postEdit successfully mitigated the impact of knowledge conflicts through post-processing. We further verify the excellent robustness of postEdit for base LLM output formats and architectures in Appendix D.3 and D.4.

## 6.2 Case Study

To visually demonstrate the editing and style retention of postEdit and baselines, we conduct the case study in Tab. 4. In Case 1, postEdit accurately identifies and modifies "*French*" to "*Italian*" while maintaining the rest of the text unchanged to keep the style to the greatest extent. In contrast, IKE only responds with "*Italian*" and SERAC replies with "*Marcel Maupi was Italian*" without referencing the original response, revealing serious style over-editing. In Cases 2 and 3, postEdit respectively replaces "*FIFA (Fédération Internationale de Football Association)*" with "*Avengers*" and modifies "*Common Lisp*" to "*JavaScript*". This demonstrates that postEdit can locate and edit spans semantically related to editing knowledge, going beyond a rudimentary replacement of old objects with new ones. Furthermore, it is evident that postEdit can handle spans logically associated with the editing. In Case 4, the location changes from "*Antarctica*" to "*Europe*", and the span in the original response, describing the location as "*the northern part of the Antarctic Peninsula*", is correspondingly adjusted to "*the northern part of the continent*". Similarly, in Case 5, as "*Manchester*" is changed to "*Bilbao*", the country is also edited from "*England*" to "*Spain*".

## 6.3 Ablation Study

***Module Ablation.*** To understand each component's role in postEdit, we conduct ablation study in Tab. 5. In our postEdit framework, we utilize GPT-4 to generate edited responses and subsequently perform data filtering. After removing data filtering, the SE score for INS queries exhibits a decline (Simple -1.9 and Rephrase -1.5), indicating that data filtering effectively enhances the quality of training data. Replacing the post-editor with ChatGPT results in a noticeable decline in performance across different types. This suggests that LLMs like ChatGPT are not proficient performing such editing tasks, highlighting the need for fine-tuning the post-editor. Substituting GPT-4 with ChatGPT for edited response augmentation results in a slight SE score increase (avg +0.13) but a significant SR score decrease (avg -2.78). This indicates that ChatGPT lacks the fine-grained granularity in editing compared to GPT-4, thereby resulting in a coarser-grained post-editor. Finally, we introduce

**Table 5: Ablation Study on CounterFact.**

| Method | Semantic Editing (SE) | | | | Semantic Retention (SR) | | | |
|---|---|---|---|---|---|---|---|---|
| | Simple | Rephrase | OOS | AVG | Simple | Rephrase | OOS | AVG |
| postEdit | 92.5 | 92.1 | 99.4 | **94.67** | 93.9 | 94.02 | 99.82 | **95.91** |
| *Module Ablation* | | | | | | | | |
| -w/o data fillter | 90.6 | 90.6 | 99.4 | 93.53 | 94.19 | 93.76 | 99.82 | 95.92 |
| post-editor→ChatGPT | 89.73 | 87.8 | 70.77 | 82.54 | 89.39 | 88.78 | 83.27 | 86.26 |
| GPT4→ChatGPT | 93.2 | 91.8 | 99.4 | 94.80 | 90.04 | 89.54 | 99.81 | 93.13 |
| SBERT Judgement | 92.2 | 85.2 | 96.3 | 91.23 | 94.47 | 92.49 | 98.97 | 95.31 |
| *Training Data Ablation* | | | | | | | | |
| -w/o Simple | 91.8 | 91.2 | 99.5 | 94.17 | 93.96 | 94.21 | 99.89 | 96.02 |
| -w/o Rephrase | 92 | 12.9 | 99.8 | 68.23 | 94.37 | 71.67 | 99.95 | 88.66 |
| -w/o OOS | 92.2 | 91.5 | 4.7 | 62.8 | 94.47 | 94.12 | 75.01 | 87.86 |

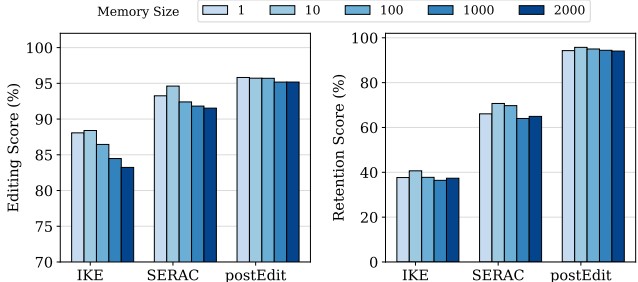

**Figure 5: Performance of methods under different Edit Memory size on CounterFact.**

the editing judging module, the same as SERAC, through comparing the SBERT semantic similarity with a threshold. The observed decrease in Rephrase and OOS scores demonstrates the superior discriminative capability of the post-editor.

***Training Data Ablation.*** To understand the role of each training data type in postEdit, we further conduct data ablation by removing each type of data from the training set. In Tab. 5, we observe that removing Simple data has no notable impact, while the removal of Rephrase data leads to a significant drop (-79.2) in the SE metric. This indicates that Rephrase data plays a crucial role in improving the post-editor's ability for editing knowledge injection and generalization, while relying solely on Simple data doesn't suffice for achieving the post-editor's generalization. After removing OOS data, although there is a noticeable decline in OOS metrics, the metrics for Simple and Rephrase do not show a discernible improvement. This indicates that post-editor doesn't excessively compromise its ability to perform edits when learning to discriminate editing.

### 6.4 Effect of Memory Size

In real-world scenarios, as the world evolves, edited knowledge should be continuously infused and preserved, i.e., the size of Edit Memory will continue to expand[9]. For the edit retrieved from Edit Memory, IKE utilizes the base LLM itself, SERAC applies a similarity threshold, and postEdit employs the post-editor to determine whether the query is within the scope of editing. We evaluate the

---
[9]In some studies, this corresponds to Batch Editing and Sequence Editing.

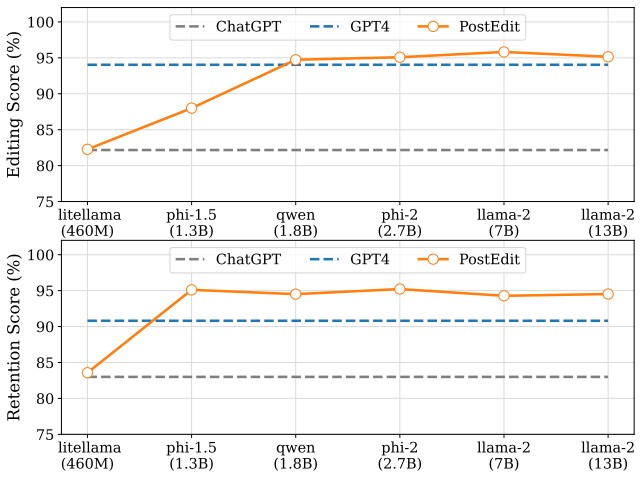

**Figure 6: Performance curves of the post-editor at different scales on CounterFact.**

performance of these methods under varying memory sizes in Fig. 5. With the same retriever, postEdit exhibits the highest robustness among methods in both Editing and Retention scores, substantiating the superiority of the postEdit mechanism in discerning the necessity of editing.

### 6.5 Effect of Post-editor Scale

To investigate the effect of post-editor scale on performance, we compare evaluation scores across models ranging from 460M to 13B in size. As illustrated in Fig. 6, it is evident that with the increase in post-editor scale, editing scores gradually improve (significant from 460M to 1.8B, followed by slower gains beyond 1.8B), while retention score remains stable after reaching 1.3B. This suggests that editing ability is more influenced by the model scale, and a larger post-editor can enhance editing performance while maintaining the retention. We also compare the effectiveness of post-editor with zero-shot ChatGPT and GPT-4. Similar to the findings in Section 6.3, LLMs like ChatGPT are not proficient in executing the editing task. Therefore, on CounterFact, the performance of the 460M post-editor is comparable to ChatGPT, and the 1.8B post-editor surpasses GPT-4. This indicates that the postEdit framework does not rely on a large-scale post-editor, and small-sized editors can achieve satisfactory performance and high efficiency.

### 7 Conclusion

In this paper, we firstly introduce a comprehensive evaluation framework for knowledge editing under black-box LLMs, incorporating multiple perspectives and considering the style retention. Next, we propose a novel postEdit framework to address existing issues in privacy leakage of editing data and style over-editing in current methods by post-processing the output of LLMs. Finally, experiments on two benchmarks and thorough analysis demonstrate that postEdit outperforms all baselines and achieves strong generalization.

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

# A Details of Evaluation

## A.1 Details of Existing Metrics

There are three metrics based on logits mainly used to evaluate the performance of knowledge editing in previous work, namely Efficacy, Generalization, and Specificity.

- **Efficacy** measures the accuracy of knowledge editing using **ES** (Efficacy Score) and **EM** (Efficacy Magnitude). For Simple type queries, the meaning of ES is $E[I[P(o^*) > P(o)]]$, and EM is obtained by $E[P(o^*) - P(o)]$.

- **Generalization** measures the accuracy of knowledge editing on Rephrase queries by using **RS** (Rephrase Score) and **RM** (Rephrase Magnitude). For Rephrase type queries, RS and RM are actually calculated to derive ES and EM under the condition of rephrasing queries.

- **Specificity** uses **NS** (Neighborhood Score) and **NM** (Neighborhood Magnitude) to measure the ability of knowledge editing to preserve unrelated knowledge. When dealing with OOS queries beyond the editing scope, no editing should take place, and the original facts should be preserved. Therefore, NS is obtained by $E[I[P(o) > P(o^*)]]$, and NM is obtained by $E[P(o) - P(o^*)]$.

## A.2 Pseudo-code of Evaluation Framework

We summarize the pseudo-code of our proposed evaluation framework in Algorithm 1.

## A.3 Consistency with Human Evaluation

In Section 3.3, we proposed a comprehensive evaluation framework, incorporating editing metrics (TE, SE) and retention metrics (TR, SR) to evaluate the quality of output text after knowledge editing. Prior to employing these metrics for evaluation, it was imperative to ensure their validity and necessity. To address this, we sample 300 data points from the test set (comprising Simple, Rephrase, and

**Table 6: The Pearson correlation coefficient between auto metrics and manual scores. For the auto metrics, Editing is the average of TE and SE; Retention is the average of TR and SR; Overall is the average of Editing and Retention.**

| Human Score | Auto Metric | Pearson Correlation |
|---|---|---|
| Editing | TE | 0.7644 |
| | SE | 0.7784 |
| | Editing | 0.8074 |
| Retention | TR | 0.9195 |
| | SR | 0.8868 |
| | Retention | 0.9255 |
| Overall | Editing | 0.5356 |
| | Retention | 0.7612 |
| | Overall | 0.839 |

OOS examples in a 1:1:1 ratio) and enlist human evaluators to independently score them from the perspectives of editing, retention, and overall assessment.

The rules for human scorers scoring the effectiveness of knowledge editing are as follows: in terms of editing, for INS queries, scoring is as follows: 0 points if there is no editing at all; 0.5 points if there are partial edits, and the sentence still retains old knowledge or exhibits logical inconsistencies; 1 point for perfect knowledge editing with no issues. For OOS queries, the scoring rules are reversed. In the retention aspect, after disregarding content related to the edited knowledge in the sentence, for responses within the editing scope: 0 points for very poor consistency between new and old responses; 0.5 points for ordinary consistency; 1 point for excellent consistency. In the overall aspect, human scorers are required to consider the overall impact of knowledge editing and assign scores within the range of 0, 1, 2, 3, 4 to the edited outputs. Then, we conduct Pearson correlation analyses between these human scores and our automated metrics.

As shown in Tab. 6, both textual metrics (TE, TR) and semantic metrics (SE, SR) demonstrate commendable consistency scores with human ratings, affirming the effectiveness of the proposed metrics. Moreover, Whether for editing or retention, the consistency score of the joint assessment of textual and semantic dimensions surpasses that of any individual metric. This underscores the necessity of incorporating both textual and semantic metrics in the evaluation process. Finally, the Pearson correlation coefficient between auto editing and human overall score is a mere 0.5356. However, a combined evaluation of editing and retention metrics yield a significantly higher consistency score of 0.839 with human judgments. This suggests that effective alignment with human preferences cannot rely solely on editing scores but requires a comprehensive assessment integrating both editing and retention metrics.

# B Details of Method

In this Section, we first present the pseudo-code for postEdit training and inference in B.1. Next, we list all the prompts used by postEdit in B.2. Finally, in B.3 and B.4, we ensure the fairness of the training process, avoiding data leakage and introducing bias.

### B.1 Pseudo-code of PostEdit

We summarize the pseudo-code for training post-editor and inference of postEdit in Algorithm 2 and Algorithm 3, respectively.

### B.2 Details of Prompts

We demonstrate the two prompt templates $T^{aug}$ and $T^{edit}$ used in the postEdit method as follows:

---

**Prompt Template $T^{aug}$**

For the following query and original response, you need to follow in order:
Firstly, locate all spans related to the **old fact:{s} {r} {o}** in original reply;
Secondly, modify these spans according to **new fact: {s} {r} {o*}**.
Thirdly, output the edited response based on the modified spans (Do not output other content).
### The query:
{x}
### Original response:
{$y_o$}
### Edited response:

---

**Prompt Template $T^{edit}$**

### Instruction:
You will assume the role of an editor. For the following query and original response, if the new fact impacts the query or original response, incorporate the new fact into the original response. If not, simply output the following word: retain.
### New fact:
The answer of {s} {r} has been updated from {o} to {o*}.
### The query:
{x}
### Original response:
{$y_o$}
### Edited response:

---

### B.3 Does Training Data Cause Data Leakage for Testing?

In the experiment setup of KE, **the edits in the training set and the test set are completely non-overlapping**. Therefore, the post-editor can not rely on edits seen during training for testing. To further investigate this, we conduct an experiment as shown in Tab. 7, where we test postEdit's performance on test set samples without passing any editing information to the post-editor. If some of the editing knowledge used for testing leaks during training, postEdit successfully edits a portion of the INS test samples. However, the editing success rates for both Simple and Rephrase types are (approximately) 0% on both datasets, thereby proving that no potential data leakage occurs. This also demonstrates that post-editor relies on edit knowledge guidance for INS/OOS judgment

**Table 7: Test results for CounterFact and zsRE when Edit Memory is empty. We simulate this scenario by replacing the recalled edit with an empty string "".**

| Types | CounterFact | | zsRE | |
|---|---|---|---|---|
| | TE | SE | TE | SE |
| Simple | 0.0 | 0.0 | 0.0 | 0.67 |
| Rephrase | 0.0 | 0.0 | 0.0 | 0.33 |
| OOS | 100.0 | 98.59 | 100.0 | 100.0 |
| AVG | 33.33 | 32.86 | 33.33 | 33.67 |

and revisions, rather than memorizing patterns from the training data.

### B.4 Does Using GPT for Data Augmentation Introduce Bias?

In the training stage, we incorporate data augmentation from both ChatGPT and GPT-4 to construct high-quality editing training data. Although GPT models are well-aligned, we further detect and address potential data bias through the following two aspects:

- **Bias in generation quality.** We perform data filtering based on TE and SE metrics for the generated data by GPT-4, discarding low-quality biased data, as shown in Section 4.2 (Edited Response Augmentation).

- **Bias in ethics and safety.** We use the LlamaGuard model [14] to evaluate the generated content. Since the datasets used in this work are knowledge-based rather than related to sensitive fields like safety and ethics, we achieve a **100% safety judgment** result. This demonstrates that our approach does not introduce ethical or bias issues.

## C Details of Experiments Setup

In this section, we provide detailed descriptions of the experimental datasets, baselines, and implementation processes in Appendix C.1, C.2, and C.3, respectively.

### C.1 Details of Datasets

In this work, we mainly used two datasets: zsRE and CounterFact.

- **zsRE** [17] is one of the most popular question answering (QA) datasets which use question rephrasing as the equivalence neighborhood. These queries of Rephrase type are generated by back-translation. In zsRE, the relationship between entities is associated with a set of crowd-sourced generated questions. Additionally, zsRE associates questions with randomly generated sentences to add out-of-editing scope examples.

- **CounterFact** [23] is a more challenging dataset than zsRE, the expected output of which is contradictory to the fact. It is built to distinguish superficial alterations in the word selections and significant, generalized modifications in its foundational factual knowledge. In CounterFact, the edited answer to the question can sometimes be counterfactual to real world, which makes it harder for the model to predict desired answer and avoid the

**Table 8: Statistical information on the sampled datasets.**

| Dataset | Data Type | Train Number | Test Number | Length of Original Response (mean/max) |
|---|---|---|---|---|
| CounterFact | ALL | 30000 | 1500 | 51.34/436 |
| | Simple | 10000 | 500 | 50.40/436 |
| | Rephrase | 10000 | 500 | 53.03/374 |
| | OOS | 10000 | 500 | 50.59/367 |
| zsRE | ALL | 30000 | 1500 | 22.39/406 |
| | Simple | 10000 | 500 | 14.84/119 |
| | Rephrase | 10000 | 500 | 18.38/257 |
| | OOS | 10000 | 500 | 33.96/406 |

effects of pre-trained LLMs knowing these desired facts before editing.

Following the previous work [42], for CounterFact, we designate data with edit id numbers ranging from 0 to 2000 as the test set for knowledge edit, while the remaining data constitute the training set. As we adopt ChatGPT as our base LLM in main experiments, in order to control the dataset size, we randomly sampled 30,000 examples (10,000 each for Simple, Rephrase, and OOS) from the original training set. These samples constitute our training set. Additionally, we randomly selected 1,500 examples (500 each for Simple, Rephrase, and OOS) from the original test set to create our query test set. The original response for INS test queries are ensured to hit the old knowledge object before editing, and the OOS are ensured to have no wrong knowledge before editing. We present the statistical information of the datasets after sampling in Tab. 8, and show a training sample and test sample from zsRE respectively as follows:

---

**Sample From zsRE Training Set**

```
{
    "edit_id": 15000,
    "edit": "Denis Dyack » Denys de La Tour || Who is the
designer of Too Human?",
    "query": "Who is the designer from Too Human?",
    "query_type": "rephrase",
    "original_response_by_gpt3.5": "The designer of
Too Human is Denis Dyack.",
    "edited_response_by_gpt4": "The designer of Too
Human is Denys de La Tour." }
```

---

**Sample From zsRE Test Set**

```
{
    "edit_id": 70,
    "edit": "Serpens » Andromeda || Which constellation
is NGC 6604 in?",
    "query": "Which constellation does NGC 6604 belong
to?",
    "query_type": "rephrase",
```

---

```
    "original_response": "NGC 6604 belongs to the con-
stellation of Serpens."
}
```

## C.2 Details of Baselines

- **IKE** [42] is a method of knowledge editing that does not involve modifying the parameters of LLMs. It defines three types of demonstration formatting templates including copy, update, and retain. These templates serve distinct functions and act as guiding principles for the language model, enabling it to edit knowledge through in-context learning, allowing IKE to maintain both efficiency and excellent generalization and specificity. This opens up the possibility of employing IKE for the task of knowledge editing even in scenarios involving black-box models.

- **PROMPT** [42] is similar to IKE, as a method of knowledge editing through in-context learning. However, unlike IKE, PROMPT doesn't require constructing three types of demonstrations but directly provides new knowledge to the LLM for knowledge editing.

- **SERAC** [28] is a memory-based method of knowledge editing. This method stores edits in explicit memory and learns to reason about these edits as needed to adjust the predictions of the base LLM without modifying parameters. SERAC uses an explicit cache of user-provided edit descriptors, alongside a scope classifier and surrogate model. When presented with a query, SERAC uses the scope classifier to determine if the query falls within the editing scope. If it does, the output is predicted via the surrogate model; otherwise, it defers to the base LLM for the output.

- **SERAC (ChatGPT)** In SERAC, the surrogate model is obtained by fine-tuning a smaller language model compared to the base LLM. We utilize ChatGPT as the surrogate model to derive a SERAC variant that requires no additional training.

## C.3 Details of Implementation

As described in Section 3.3, our evaluation framework employs a NLI model for computing SE, ROUGE scores for computing TR, and a SBERT model for computing SR. In details, SE utilizes albert-xxlarge-v2-snli_mnli_fever_anli_R1_R2_R3-nli[10] as the NLI model;

---

[10]https://huggingface.co/ynie/albert-xxlarge-v2-snli_mnli_fever_anli_R1_R2_R3-nli

**Table 9: Performance comparison on CounterFact.**

| Method | Textual Editing (TE) | | | | Semantic Editing (SE) | | | | Textual Retention (TR) | | | | Semantic Retention (SR) | | | |
|---|---|---|---|---|---|---|---|---|---|---|---|---|---|---|---|---|
| | Simple | Rephrase | OOS | AVG (HM) | Simple | Rephrase | OOS | AVG (HM) | Simple | Rephrase | OOS | AVG (HM) | Simple | Rephrase | OOS | AVG (HM) |
| MeLLo | 42.42 | 32.87 | 37.07 | 37.55 (37.05) | 43.61 | 35.11 | 44.3 | 41.11 (40.55) | 16.42 | 11.22 | 15.59 | 14.47 (14.01) | 38.5 | 31.61 | 41.58 | 37.32 (36.74) |
| DeepEdit | 47.0 | 40.0 | 27.03 | 38.03 (36.03) | 52.2 | 44.57 | 39.02 | 45.31 (44.63) | 19.51 | 16.22 | 15.65 | 17.16 (16.97) | 39.24 | 35.41 | 39.14 | 37.97 (37.84) |
| RARR | 53.9 | 49.47 | 85.67 | 63.17 (59.48) | 55.9 | 50.96 | 86.48 | 64.6 (61.13) | 54.18 | 54.9 | 63.19 | 57.44 (57.15) | 62 | 62.98 | 71.13 | 65.39 (65.12) |
| RAG-8shot | 99.7 | 99.79 | 9.35 | 69.32 (23.62) | 98.9 | 95.64 | 11.79 | 68.54 (28.47) | 26.2 | 23.98 | 4.57 | 18.21 (10.04) | 55.32 | 53.5 | 25.01 | 44.54 (39.09) |
| postEdit (ours) | 96.8 | 94.7 | 99.4 | 96.97 (96.93) | 92.5 | 92.1 | 99.4 | 94.67 (94.55) | 88.65 | 89.66 | 99.64 | 92.65 (92.39) | 93.9 | 94.02 | 99.82 | 95.91 (95.84) |

ROUGE score is implemented through the rouge library[11], using the F1 score of ROUGE-1; SR uses all-MiniLM-L6-v2[12] as the SBERT model.

For training of post-editor, we employ ChatGPT (gpt-3.5-turbo-0301) for original response augment and GPT-4 (gpt-4-0613) for edited response augment [13], with the default temperature coefficient ($t = 0.1$). In order to enhance training efficiency and reduce the number of updated parameters, we adopt the LoRA strategy [12] to finetune LLaMA 2-7B. Specifically, the rank of LoRA is set to 8, with *lora_alpha* at 16 and *lora_dropout* at 0.05. The LoRA update matrix is applied to the self-attention and FFN layers, with *target_modules* as ["q_proj","k_proj","v_proj","o_proj","gate_proj", "down_proj","up_proj"]. We train 5 epochs to optimize post-editor, employing a batch size of 128 and a learning rate of 5e-2. We also use the warmup and cosine annealing strategy, with a warmup ratio of 0.1 and the Adam optimizer [16].

For retriever of postEdit, consistent with all baselines, we use all-MiniLM-L6-v2 to encode queries and edit knowledge, while employing dot product as the similarity function. For base LLM, we use ChatGPT (gpt-3.5-turbo-0301) in main experiments, with a temperature coefficient of 0.1. During inference of post-editor, we set the temperature coefficient of 0.1 and use beam search to decode the output, where *num_beams* is set to 4. To further improve the inference speed, we apply 8-bit quantization when loading post-editor.

In terms of baselines, for SERAC, we fine-tune the surrogate model using the same LLAMA2-7B as post-editor and the similarity discrimination threshold is set at 0.7, determined through hyperparameter search on the training set (ranging from 0.1 to 0.9 with a step size of 0.1). To better maintain consistency between baselines and postEdit implementations, we adopt training output targets consistent with postEdit for the surrogate model of SERAC, i.e., GPT-4 augmented edited response, rather than new objects of editing knowledge, aiming to achieve higher stylistic retention. For IKE, we set the number of demonstration examples to 32. The rest of the hyperparameter settings for the baselines follow the default configurations in their original papers. All experiments use a single Nvidia A100 GPU (80 GB of memory).

## D More Experiments

In this section, we compare postEdit with other task baselines in D.1. In D.2, we investigate postEdit's performance in multi-hop reasoning scenarios regarding edited knowledge. In D.3 and D.4, we

further verify postEdit's robustness across different output formats and architectures of the base LLM. Finally, in **??**, we conduct ablation experiments on the training data to thoroughly examine postEdit.

### D.1 Comparison with More Baselines

In Section 5, we compared methods that have the same scenario as postEdit. For a comprehensive comparison, we transfer some methods from other task scenarios as baselines to further enrich the experiments:

- MeLLo [43] is a method specifically designed for multi-hop reasoning scenarios in knowledge editing, storing edited facts externally and iteratively prompts LLMs to generate answers consistent with the edited facts.

- DeepEdit [35] designs decoding constraints to "regulate" LLMs' reasoning, enhancing logical coherence when incorporating new knowledge for scenarios requiring multi-hop reasoning regarding edited knowledge.

- RARR [9] aims to reduce hallucinations in LLM outputs by scrutinizing and revising. It initially uses search engines for evidence and attribution, then corrects unsupported content while preserving the original output, achieved through few-shot demonstrations. We replace the search engine with edit memory.

- In addition to PROMPT and IKE, similar to the conventional RAG approach, we utilize few-shot <query, edit, edited output> prompts to enhance the base LLM's utilization of editing knowledge, where all demonstration samples belong to the INS type, referred to as RAG-8shot. [14]

The results are shown in Tab. 9. Overall, postEdit still outperforms all baselines. We can further observe that: Firstly, since MeLLo, DeepEdit, and RARR are not designed specifically for general knowledge editing scenarios, they perform poorly on CounterFact. Secondly, leveraging the impressive in-context learning capabilities of ChatGPT, RAG-8shot achieves near-perfect INS Editing scores, but faces significant challenges on OOS Editing due to the lack of OOS demonstrations. This emphasizes the need for a INS/OOS judgment mechanism on top of RAG. Lastly, post-processing methods (postEdit, RARR) achieve higher Retention scores compared to pre-processing methods (MeLLo, RAG-8shot), highlighting the advantage of post-processing for style retention.

---

[11]https://pypi.org/project/rouge
[12]https://huggingface.co/sentence-transformers/all-MiniLM-L6-v2
[13]https://platform.openai.com/docs/models

[14]Since in the standard KE experimental setup, the size of edit memory is set to 1, serving as an "oracle" retrieval setting to encourage methods to focus more on editing and locality capabilities. Therefore, we don't compare with some RAG methods that focus on improving retrieval recall.

**Table 10: An example of postEdit on MQuAKE-CF-3K.**

| Type | Content |
|------|---------|
| Edits | 1. Lalu Prasad Yadav is married to Rabri Devi → Victoria Nuland
2. Victoria Nuland is a citizen of United States of America → Germany
3. Germany is located in the continent of Europe → Antarctica |
| Query | What continent is the country of citizenship of the spouse of the founder of the Rashtriya Janata Dal party located in? |
| Original response | The founder of the Rashtriya Janata Dal (RJD) party is Lalu Prasad Yadav. His spouse, Rabri Devi, is a citizen of India. India is located on the continent of Asia. |
| Edited response | The founder of the Rashtriya Janata Dal (RJD) party is Lalu Prasad Yadav. His spouse, Victoria Nuland, is a citizen of Germany. Germany is located on the continent of Antarctica. |

**Table 11: The performance of methods under a single group of edits in MQuAKE-CF-3K.**

| Method | MeLLo | DeepEdit | PostEdit |
|--------|-------|----------|----------|
| ACC | 35.8 | 61.00 | 64.26 |

**Table 12: Performance of postEdit in different original output formats on CounterFact.**

| Format | TE | SE | TR | SR |
|--------|-----|-----|-----|-----|
| Origin | 96.97 | 94.67 | 92.65 | 95.91 |
| Simplification | 96.45 | 94.95 | 93.43 | 96.53 |
| Verbose | 92.07 | 91.57 | 96.17 | 96.9 |
| Humor | 95.56 | 93.2 | 97.42 | 97.28 |

## D.2 Performance of PostEdit in Multi-hop Knowledge Editing

It is important to emphasize that this paper primarily focuses on general knowledge editing scenarios, rather than scenarios requiring multi-hop reasoning for edited knowledge . Nonetheless, more diverse scenarios are indeed beneficial for understanding postEdit. Therefore, we use the multi-hop editing dataset MQuAKE-CF-3K [43] as the test set and the remain in MQuAKE-CF as the training set. To solve the scenario where one query might correspond to multiple edits in MQuAKE, we modify Prompt $T^{aug}$ and $T^{edit}$ (Appendix B.2) to accommodate multiple edits. We test under single-group editing and follow DeepEdit's settings (testing only the first question of each instance and evaluating via ACC) to align with baselines.

It can be seen in Tab. 11 that under a single-group of edits, postEdit still outperforms the baselines. As shown in Section 6.2, the high-quality training data enables the post-editor to handle spans logically associated to edits. It is worth emphasizing that postEdit achieves this solely through a single-pass base LLM and editor inference, unlike MeLLo and DeepEdit, which rely on iterative inference and modifications, resulting in high costs. An case of postEdit is shown in Tab. 10. However, under the multi-group editing settings in MQuAKE (e.g., 100 edits), postEdit should be coupled with a retriever proficient in multi-hop question retrieval to tackle the retrieval challenges encountered in multi-hop editing. Since improving retrieval is not the focus of this study, we omit this part and leave it for future work.

## D.3 Robustness of PostEdit to Original Response Format

Given the diverse output formats of base LLM, which may differ from the training data format, it is crucial to investigate the robustness of postEdit for different original output formats during inference. To explore this, we utilize GPT-4 to rewrite the original output (Origin) of the base LLM, including more concise outputs (Simplification), more verbose outputs (Verbose), and outputs presented in a humorous manner (Humor). Since the rewrites are only done during the testing phase, they can be considered out-of-domain formats for the training data. The experimental results on CounterFact are shown in Tab. 12. It can be seen that the three output variants do not significantly affect the results, demonstrating postEdit's robustness to different output formats. The worst performance is under the Verbose format, primarily because the longer original output poses a higher challenge to the post-editor, resulting in slight decreases in TE and SE. However, it also led to higher style consistency.

## D.4 Robustness of PostEdit to Base LLM Architecture

Methodologically, postEdit only requires the text output from the base LLM, without needing to access any internal information of base LLM. This not only allows postEdit to be applied in black-box LLM scenarios but also decouples the editing process from the base LLM. As a result, it can be reused with different base LLMs as a downstream plugin without the need for retraining. To experimentally verify this, we present the results of directly reusing the post-editor initially trained for ChatGPT on other base LLMs in

**Table 13: Performance of PostEdit in different original output formats under CounterFact.**

| Base LLM | Architecture | TE | SE | TR | SR |
|----------|--------------|------|------|------|------|
| GPT-3.5 | Unknown | 96.97 | 94.67 | 92.65 | 95.91 |
| PaLM2 | Causal Decoder | 95.49 | 92.79 | 95.64 | 97.49 |
| Llama2-70B-chat | Causal Decoder | 95.7 | 94.73 | 91.06 | 94.78 |
| Mixtral-8×7B | Mixture of Experts | 96.25 | 94.4 | 93.94 | 96.89 |
| GLM-4 | Prefix Decoder | 93.77 | 92.08 | 97.64 | 98.17 |

Tab. 13. The results show that postEdit exhibits strong generalization across the current mainstream architectures, confirming its flexibility as a post-processing plugin for various base LLMs.

## E  Limitations

This paper primarily investigates the assessment and methodology of knowledge editing in black-box LLM scenarios. The proposed evaluation framework can comprehensively assess edited responses from multiple perspectives, and the postEdit method effectively addresses issues related to privacy concerns of editing data and style over-editing. However, our work also has several limitations: (1) Although our proposed evaluation framework and postEdit method mainly focus on knowledge editing in black-box LLM scenarios, they can be equally applied to editing in white-box LLM scenarios. Due to constraints in length and the focus of the paper, we haven't thoroughly explored this in the paper. (2) Although the postEdit framework does not require retraining when injecting editing knowledge, it still necessitates an initial fine-tuning phase to enable the post-editor to learn the ability to discern whether a query is within the editing scope and how to perform the editing, resulting in a certain computational load. (3) Our study primarily investigates the application of knowledge editing in knowledge question answering tasks, similar to previous research. We believe that our framework can be extended to other scenarios, such as fact-checking and sentiment editing. We leave these explorations for future research.

## F  Ethic Consideration

In this paper, we propose a knowledge editing approach that can be flexibly applied downstream to post-process the outputs of LLMs, effectively safeguarding the privacy of downstream private editing data and maintaining consistency in the style of the LLM. While the purpose of knowledge editing is to rectify errors or outdated knowledge in LLMs, malicious knowledge editing may lead to the generation of harmful or inappropriate outputs by the model. Therefore, ensuring secure and responsible practices in knowledge editing is of paramount importance. The application of these techniques should be guided by ethical considerations, with safeguard measures in place to prevent misuse and mitigate the potential for harmful outcomes. Additionally, due to the difficulty in obtaining continuously up-to-date knowledge, some KE datasets such as CounterFact use counterfactual knowledge to validate the effectiveness of methods. Furthermore, the base LLM, such as ChatGPT used in this work, merely serves as a demonstration of research on knowledge editing in black-box model scenarios. We emphasize that these datasets and LLMs are solely for academic exploration and do not involve actual applications in real-world scenarios, nor do they include content modification or attacks on commercially used LLMs.

Received 20 February 2007; revised 12 March 2009; accepted 5 June 2009

**Algorithm 1:** Pseudo-code of Evaluation Framework in a Python-like style.

```
# x: the input of LLM (All text is processed in lowercase, the same below.)
# x_label: "INS" if x in editing scope else "OOS"
# y_o, y_e: the original and edited output of LLM
# o_old, o_new: the object of old knowledge t and new knowledge t* for editing
# k_old, k_new: text format of t and t*
# k_self: text format of LLM's self-knowledge t_o and is equivalent to [x, y_o]
# func_entail(a,b): return True if a entails b else False by using a NLI model
# func_rouge(a,b): return the ROUGE socre of a and b
# func_sim(a,b): return the similarity of a and b using a SBERT model

def TE(y_e, x_label, o_old, o_new):
    ctn_old=1 if o_old in y_e else 0
    ctn_new=1 if o_new in y_e else 0
    if x_label=="INS":
        TE_score=0.5*ctn_new + 0.5*(1-ctn_old)
    else:
        TE_score=0.5*ctn_old + 0.5*(1-ctn_new)
    return TE_score

def SE(x_label, x, y_e, k_old, k_new, k_self, func_entail):
    ent_new=1 if func_entail(x+" "+y_e,k_new) else 0
    if x_label=="INS":
        ent_old=1 if func_entail(x+" "+y_e,k_old) else 0
        SE_score=0.5 * ent_new + 0.5 * (1-ent_old)
    else:
        ent_old=1 if func_entail(x+" "+y_e,k_self) else 0
        SE_score=0.5*ent_old + 0.5*(1-ent_new)
    return SE_score

def TR(x_label, y_o, y_e, o_old, o_new, func_rouge):
    if x_label=="INS":
        TR_score=func_rouge(y_o.replace(o_old,"mask"), y_e.replace(o_new,"mask"))
    else:
        TR_score=func_rouge(y_o,y_e)
    return TR_score

def SR(x_label, y_o, y_e, o_old, o_new, func_sim):
    if x_label=="INS":
        SR_score=func_sim(y_o.replace(o_old,"mask"), y_e.replace(o_new,"mask"))
    else:
        SR_score=func_sim(y_o,y_e)
    return SR_score
```

**Algorithm 2:** Train post-editor

**Data:** training dataset $D_{train} = \{(e_i, x_i)\}$

**Require:** base LLM $f_{base}$, GPT-4 $f_{gpt4}$, trainable generative model $f_{edit}$, training epoch **E**, batch size **B**

**for** $i$ in $1, \cdots, |D_{train}|$ **do**

    $y_{i,o}^{aug} = f_{base}(x_i)$                                                              ▷**Original Response Augment**

    **if** $x_i \in \mathcal{X}_e$ **then**

        $y_{i,e}^{aug} = f_{gpt4}(T^{aug}(e_i, x_i, y_{i,o}^{aug}))$                                      ▷**Edited Response Augment**

        **if** $\text{TE}(y_{i,e}^{aug}) \neq 1$ *or* $\text{SE}(y_{i,e}^{aug}) \neq 1$ **then**

            **delete** $(e_i, x_i, y_{i,o}^{aug}, y_{i,e}^{aug})$

        **end**

    **else**

        $y_{i,e}^{aug} = \langle Retain \rangle$

    **end**

**end**

$D_{train}^{aug} = \{(e_i, x_i, y_{i,o}^{aug}, y_{i,e}^{aug})\}$

**for** $epoch$ in $1, \cdots, E$ **do**

    **for** $iter$ =0, 1, 2, $\cdots$ **do**

        sample a mini-batch **B** from $D_{train}^{aug}$                                      ▷**Supervised Fine-tuning**

        compute $\mathcal{L}_{sft}$ by equation 6 and optimize $f_{edit}$

    **end**

**end**

**Output:** trained post-editor $f_{edit}$

---

**Algorithm 3:** Inference of PostEdit

**Input:** use query $x$

**Require:** Edit Memory $M_e$, base LLM $f_{base}$, post-editor $f_{edit}$, SBERT retriever $f_{retr}$

get original response: $y_o = f_{base}(x)$

retrieve the most similar edit index: $i^* = \text{argmax}_{0 \leq i < |M_e|} \ \text{sim}(x, e_i)$

get post-editor's output: $f_{edit}(x_{edit}) = f_{edit}(T^{edit}(e_{i^*}, x, y_o))$

**if** $f_{edit}(x_{edit}) \neq \langle Retain \rangle$ **then**

    $y_e = f_{edit}(x_{edit})$

**else**

    $y_e = y_o$

**end**

**Output:** final response $y_e$

