# OpenReview forum: "Assessing and Post-Processing Black Box Large Language Models for Knowledge Editing"
_ACM.org/TheWebConf/2025/Conference — WWW 2025 Oral_

### Official Review · Reviewer_b1K8 · 2024-11-30

**Novelty:** 5
**Technical Quality:** 5

**Review:**

This paper introduces a method for knowledge editing for black-box LLMs that addresses previous limitations related to privacy leakage and style over-writing. Additionally, two evaluation methods are proposed to assess both editing and retention.

Pros:

1.	The experimental section of this paper is comprehensive and detailed, effectively illustrating all the relevant elements mentioned in the text through experimentation.

2.	The paper provides a comprehensive description of the problem at hand and the methods used to address it, making it easily understandable for readers.

Cons:

1.	The paper outlines the drawbacks of recent work but fails to explain how these shortcomings such as style retention play a crucial role in the knowledge editing task and therefore need to be addressed.

Other comments:

1.	Typo in Figure2 (b) SERAC OOS(out of the editing scope) should be input to black box LLM while INS to surrogate LM

2.	Typo in Table5: data filter

**Questions:**

Could you please provide a brief explanation on the significance of retaining the retention of responses? It seems that for QA tasks, ensuring correctness in responses is already enough.

**Reviewer Confidence:**

2: The reviewer is willing to defend the evaluation, but it is likely that the reviewer did not understand parts of the paper

**Scope:**

4: The work is relevant to the Web and to the track, and is of broad interest to the community

---

### Official Review · Reviewer_7UcN · 2024-11-30

**Novelty:** 3
**Technical Quality:** 4

**Review:**

The paper introduces a  framework, postEdit, designed for the knowledge editing (KE) of black-box LLMs. Utilizing the online LLM as the first training data and original result of LLM response. And fintuning a small LM for answer format limitation and knowledge editing. The framework aims to maintain the privacy and stylistic consistency of the edited output, leveraging a retrieval mechanism and a specially trained editing model to handle the KE tasks. Extensive experiments demonstrate that postEdit significantly outperforms existing methods in both editing efficacy and style retention.

Pro:

1.	Incorporating retention into the evaluation is an excellent idea presented for the first time in this paper. It proposes a novel evaluation framework that assesses both the accuracy of the edits and the stylistic retention, essential for maintaining the coherence and usability of language model outputs.

2.	The experimental results are robust, showing clear improvements over baseline methods across several metrics and datasets, which underscores the effectiveness of the proposed approach. In addition to performance comparison, the authors also conduct ablation studies, compatibility studies, consistency analysis, and evaluations of the answering capabilities.

Cons:

1.	The model design is relatively simple. In the existing models, the first half of the PostEdit involves using an online RAG language model to gather basic information. The second half (post-editor), which is the focus of this paper, involves fine-tuning a language model to revise knowledge and generate correct answers in the required format.

Question and suggestion: However, existing language models like GPT-4 already support structured response outputs (such as JSON, HTML) through prompting, and there are existing works exploring structured output generation. Would your work be benefit from integrating these existing approaches into your model for comparison, to enhance adaptability or complexity?

2.	The experimental design lacks comprehensive consideration. For instance, as introduced in Cons 1, the baseline methods used for comparison do not optimize for retention, which might make the direct comparisons somewhat unfair.

Question and suggestion: Could you consider using a baseline enhanced with prompt-based retention methods for a more equitable comparison? Additionally, in the ablation study, comparing the PostEdit+ post-editor model with PostEdit + a prompt-based retention method could provide more insightful results.

3.	Only two benchmarks are evaluated, which may not be sufficient to fully validate the model’s effectiveness. Expanding the evaluation to include more benchmarks could provide a deeper understanding of the model’s capabilities and limitations.

**Questions:**

1.	The article repeatedly mentions the need to separate editing information from the black-box language model due to privacy concerns. However, in the data filtering step, both edited data and unchanged answers are fed into GPT-4. Doesn't this contradict the initial intention of maintaining privacy by keeping the editing process separate from the language model?
2.	Introducing retention into the evaluation process is a thoughtful consideration, but it is essential to contemplate whether retention is genuinely beneficial or necessary in the context of knowledge editing. The primary focus should be on the effectiveness of the edits. Is retention truly valuable, or could it be easily achieved through simple plug-in methods?
3.	See above con1 &con2

**Reviewer Confidence:**

3: The reviewer is confident but not certain that the evaluation is correct

**Scope:**

3: The work is somewhat relevant to the Web and to the track, and is of narrow interest to a sub-community

---

### Official Review · Reviewer_MBkG · 2024-12-02

**Novelty:** 6
**Technical Quality:** 6

**Review:**

The authors introduce a novel framework called "postEdit," which effectively updates specific knowledge within LLMs while mitigating negative impacts on unrelated knowledge, maintaining model relevance and accuracy despite the rapid evolution of web content. The postEdit framework features a retrieval mechanism for knowledge updating and a specialized "post-editor" to ensure privacy and maintain textual style consistency.

Detailed comments:
1. This paper is well-motivated and well-written, making it easy to follow. The comparison figures are particularly clear.
2. The experiments conducted are detailed, and the performance generally surpasses the baselines, demonstrating significant advantages.
3. In addition to comparing corpora resources, it would be better to include a comparison of computational resource consumption.
4. In Table 5, the use of bold text might be incorrect. In the Semantic Editing scenario, the average scores for GPT-4 to ChatGPT are the highest.

**Questions:**

From the case study, it can be seen that some of the edits in the baseline did not yield all negative results. Have you considered a more in-depth evaluation? For example, what impact does the current edit mechanism have on alignment

**Reviewer Confidence:**

3: The reviewer is confident but not certain that the evaluation is correct

**Scope:**

4: The work is relevant to the Web and to the track, and is of broad interest to the community

---

### Official Review · Reviewer_B6AZ · 2024-12-02

**Novelty:** 5
**Technical Quality:** 4

**Review:**

This paper addresses the challenge of knowledge editing in black-box Large Language Models (LLMs), where internal model parameters are inaccessible, and only textual outputs can be modified. The authors propose a novel framework, postEdit, that introduces a retrieval mechanism and a purpose-trained post-editor to perform fine-grained, style-preserving modifications. This work is significant in maintaining the relevance and accuracy of black-box LLMs while addressing privacy concerns and minimizing unintended side effects of editing. Experimental results demonstrate that postEdit outperforms existing baselines across multiple benchmarks, particularly excelling in style retention and editing efficacy.

Strengths:

PostEdit focus on post-processing ensures robust privacy protection by avoiding access to model internals, making it ideal for third-party API-based scenarios. Additionally, its fine-grained editing approach preserves the original style of text while implementing necessary changes, as evidenced by significant improvements in style retention metrics. The method’s plug-and-play design further enhances its generalizability and extensibility, allowing it to adapt seamlessly to different LLMs and tasks while maintaining stability even when the underlying model changes. Finally, its ease of deployment, requiring no modifications to the base model, makes it highly suitable for resource-constrained environments.

Weakness:

1. Limited Contextual Understanding: As a post-processing mechanism, postEdit operates only on the generated output and may struggle to fully capture deeper semantic relationships in complex contexts. If the underlying LLM produces logically flawed outputs, postEdit might not be able to completely rectify these errors.

2. Limited Generalization to Novel Knowledge: Although postEdit excels at retaining style, its generalization to entirely novel or complex editing tasks remains uncertain. Tasks involving multi-hop reasoning or intricate edits may result in diminished performance.

3. Dependence on Retriever and Memory Modules: The retriever and memory modules require efficient handling of large-scale editing data. As the memory size grows, storage costs and retrieval performance may become bottlenecks.

**Questions:**

1. How does the postEdit method handle cases where the original output contains logical errors or lacks coherence? Could you provide additional examples or a detailed analysis of such scenarios to illustrate the method's limitations and potential mitigation strategies?

2. How does postEdit handle ambiguous or conflicting knowledge edits where the editing scope is unclear?

3. While smaller post-editors (e.g., 7B parameters) perform reasonably well, what is the impact on computational cost as model size increases? Could you provide an ablation study detailing the trade-off between editing quality and efficiency across a wider range of post-editor scales?

4. Can the postEdit method generalize to completely novel or complex multi-hop edits? If possible, provide examples or additional experiments on datasets requiring such capabilities to clarify its robustness in handling challenging tasks.

**Reviewer Confidence:**

3: The reviewer is confident but not certain that the evaluation is correct

**Scope:**

3: The work is somewhat relevant to the Web and to the track, and is of narrow interest to a sub-community